# NATIVE 3D EDITING WITH FULL ATTENTION

## ABSTRACT

Instruction-guided 3D editing is a rapidly emerging field with the potential to broaden access to 3D content creation. However, existing methods face critical limitations: optimization-based approaches are prohibitively slow, while feed-forward approaches relying on multi-view 2D editing often suffer from inconsistent geometry and degraded visual quality. To address these issues, we propose a novel native 3D editing framework that directly manipulates 3D representations in a single, efficient feed-forward pass. Specifically, we create a large-scale, multi-modal dataset for instruction-guided 3D editing, covering diverse addition, deletion, and modification tasks. This dataset is meticulously curated to ensure that edited objects faithfully adhere to the instructional changes while preserving the consistency of unedited regions with the source object. Building upon this dataset, we explore two distinct conditioning strategies for our model: a conventional cross-attention mechanism and a novel 3D token concatenation approach. Our results demonstrate that token concatenation is more parameter-efficient and achieves superior performance. Extensive evaluations show that our method outperforms existing 2D-lifting approaches, setting a new benchmark in generation quality, 3D consistency, and instruction fidelity.

## 1 INTRODUCTION

3D creative editing has great potential in numerous fields, including film production, entertainment industry, and digital gaming. Recent advancements (Erkoç et al., 2025; Barda et al., 2025; Li et al., 2025a; Fang et al., 2024), such as text-to-3D editing and image-to-3D editing, have significantly reduced the manual effort required by professional 3D artists while making 3D asset editing accessible to non-professionals. These approaches typically focus on score distillation sampling (SDS) (Poole et al., 2022) to edit 2D images and lift them into 3D assets. While generating high-quality 3D objects, these optimization-based approaches are a lengthy and computationally intensive process. To address this limitation, recent studies (Erkoç et al., 2025; Barda et al., 2025) reconstruct the edited 3D assets via a feed-forward 3D reconstruction model from single or sparse images. These methods edit the images using a multi-view diffusion model to keep the consistency of edited images. However, the paradigm of multi-view editing and then restructuring the 3D model also falls short in terms of either visual quality or 3D consistency due to its reliance on 2D space for editing.

To address these challenges, we propose a native 3D editing paradigm that directly manipulates 3D objects based on textual instructions. A primary obstacle to this goal is the absence of a large-scale, high-quality dataset for instruction-guided 3D editing. To overcome this, we introduce a systematic data construction pipeline to create a comprehensive benchmark, as illustrated in Fig. 1. Specifically, for the deletion task, we curate part-level 3D assets from the Objaverse dataset (Deitke et al., 2023) to serve as source objects. By programmatically removing distinct parts from each asset, we generate the corresponding target objects, forming a large collection of source-target pairs for deletion edits. To obtain the guiding instructions, we leverage a powerful multimodal large language model (MLLM), Gemini 2.5 (Comanici et al., 2025), to generate descriptive text that accurately reflects the transformation from the source to the target object. For addition and modification tasks, we build upon existing open-source 2D editing datasets (Ye et al., 2025) that provide paired source/target images, along with corresponding instructions. We employ the image-to-3D generation model, Hunyuan3D 2.1 (Hunyuan3D et al., 2025), to lift these 2D image pairs into high-fidelity 3D source and target objects, while retaining their original instructional texts. However, we observe that the generated 3D source and target objects often exhibit inconsistencies in geometry

and appearance due to the inherent limitations of current image-to-3D generation models. To ensure data quality, we implement a rigorous manual curation process, selectively retaining only those pairs where the generated source and target objects demonstrate strong geometric and visual consistency. This two-pronged approach, combining procedural generation with careful curation, allows us to construct a novel and diverse 3D editing dataset, paving the way for data-driven native 3D editing.

Building on this dataset, we further explore effective architectural designs for instruction-guided native 3D editing. Our investigation focuses on two distinct strategies for conditioning the model on the source 3D object. The first strategy employs a classic adapter-based approach (Ye et al., 2023), integrating the source object's features into the network via cross-attention mechanisms. The second involves a more direct and parameter-efficient method: 3D token concatenation. Specifically, our model learns the editing transformation during training by processing a concatenated sequence of the source object's and noised target object's 3D tokens, guided by the corresponding instruction. This allows the full-attention network to directly model the relationship between the source, target, and instruction with minimal additional parameters. Once trained, the model can perform editing at inference time by taking only the source object and the editing instruction as input to generate the desired 3D assets within 20 seconds. We are the first to explore token concatenation as a conditioning method in the 3D domain, and our experimental results demonstrate that this strategy significantly outperforms the cross-attention approach, yielding superior results in terms of both generation quality and fidelity to the instruction.

In summary, our contributions are as follows.

- We introduce the first large-scale, multi-modal benchmark dataset specifically designed for instruction-guided native 3D editing. Our comprehensive dataset, encompassing a wide range of addition, deletion, and modification tasks, is systematically constructed by leveraging both part-level 3D assets and by lifting existing 2D edit datasets into the 3D domain.

- We propose a novel and parameter-efficient architectural design for native 3D editing. We are the first to demonstrate in the 3D domain that conditioning via direct token concatenation significantly outperforms traditional cross-attention mechanisms, achieving higher fidelity to instructions and superior generation quality with minimal additional parameters.

- We present an effective and efficient feed-forward framework for native 3D editing. Our approach directly manipulates 3D representations, bypassing the quality and consistency issues of prior multi-view editing pipelines.

## 2 RELATED WORK

**3D Reconstruction and Generation with Large Models.** Stable Diffusion (Poole et al., 2022) pioneered 3D generation work based on Score Distillation Sampling (SDS), and subsequently, numerous outstanding studies (Xu et al., 2023; Lin et al., 2023; Melas-Kyriazi et al., 2023) have leveraged SDS optimization and its variant (Chung et al., 2023; Hertz et al., 2023) for 3D reconstruction and generation. These methods yielded high-quality 3D generation but were often slow and impractical due to their reliance on per-case optimization. Meanwhile, the generative results are limited by the ability of the pre-trained 2D generative models. To mitigate these issues, the Large Reconstruction Model (LRM) (Hong et al., 2023) proposes a feed-forward model trained on large-scale datasets to generate a NeRF from single images within 5 seconds rapidly. Subsequently, LGM (Tang et al., 2024) trained a Large Multi-view Gaussian Model to reconstruct 3D Gaussians from multiview images. GRM (Xu et al., 2024b) introduces a feedforward transformer-based model to generate 3d assets from sparse-view images. In view of the powerful creative capacity of the diffusion mode (Rombach et al., 2022) and the robust generalization ability of the feed-forward model (Hong et al., 2023), many works first employ the 2D multi-view diffusion model to generate multi-view images and then reconstruct the 3D assets with an LRM to achieve text-to-3D and image-to-3D generation, as demonstrated by extensions such as InstantMesh (Xu et al., 2024a) and Instant3DiT (Li et al., 2023). Inspired by these, many 3D editing studies (Erkoç et al., 2025; Chen et al., 2024a; Qi et al., 2024) attempt first to edit multi-view images and then reconstruct the 3D assets.

**3D Editing.** With the success of 3D generative models (Liu et al., 2023; Hong et al., 2023; Li et al., 2024; Long et al., 2024; Chen et al., 2024c; Xiang et al., 2025b; Fang et al., 2025; Li et al., 2025b), 3D creative editing has been widely studied (Prabhu et al., 2023; Erkoç et al., 2025; Bar-On

et al., 2025; Fang et al., 2024). Several methods (Haque et al., 2023; Chen et al., 2024b) employ optimization-based approaches, utilizing NeRF (Mildenhall et al., 2021) or 3D Gaussian splatting (Kerbl et al., 2023) as a 3D representation, with SDS (Poole et al., 2022) serving as the loss function. For example, Instruct-NeRF2NeRF (Haque et al., 2023) iterative edits NeRF through a text-based image editing model InstructPix2Pix (Brooks et al., 2023) to edit the training dataset. Vox-E (Sella et al., 2023), Tip-Editor (Zhuang et al., 2024), DreamEditor (Zhuang et al., 2023), FocalDreamer (Li et al., 2024b) employ SDS to align a 3D representation with a text prompt to achieve local editing. While powerful, the primary drawback for all these optimization-based approaches is a lengthy and computationally intensive per-edit process. To accelerate editing, other works propose faster, feed-forward solutions. PrEditor3D (Erkoç et al., 2025) employs multi-view image editing through a user-provided 2D mask. MVEdit (Chen et al., 2024a) denoises multi-view images jointly, then reconstructs a textured mesh from modified multi-view images. Talior3D (Qi et al., 2024) adopts editable dual-sided images to reconstruct the 3D mesh. Instant3Dit (Barda et al., 2025) employs a multiview inpainting diffusion model to modify the 3D mask region, and then reconstructs the 3D model using the Large Reconstruction Model. However, the paradigm of multi-view editing and then restructuring the 3D model also falls short in terms of either visual quality or 3D consistency. Instead of editing 3D assets from multiple views, we train a native 3D edited model with full attention, aiming to eliminate inconsistencies in overlapping areas that occur when editing individual views.

## 3 METHOD

### 3.1 PRELIMINARIES

**Recified Flow Models.** Rectified flow models formulate the generative process as learning a deterministic flow field that transports samples from a simple prior distribution (e.g., Gaussian noise $\epsilon$) to a complex data distribution (represented by samples $x_0$). This is achieved by defining a straight-line trajectory between noise and data: $x(t) = (1 - t)x_0 + t\epsilon$, where $t \in [0, 1]$. The model then learns a velocity field $v_\theta(x, t)$ that approximates the true vector field $v(x, t) = \epsilon - x_0$ along this path. The network $v_\theta$ is optimized using the Conditional Flow Matching (CFM) loss, which minimizes the discrepancy between the predicted and ground-truth velocity vectors:

$$\mathcal{L}_{CFM}(\theta) = \mathbb{E}_{t, x_0, \epsilon} \left[ ||v_\theta(x, t) - (\epsilon - x_0)||_2^2 \right]. \tag{1}$$

**Structured 3D Diffusion Models.** Our work builds upon a pre-trained structured 3D latent diffusion model (Xiang et al., 2025b) that employs a two-stage pipeline to create structured 3D latents. This approach disentangles the generation of coarse geometry from fine-grained details, enhancing scalability and efficiency. The generation process consists of two main stages: Sparse Structure Generation and Local Latent Generation. In the first stage, a transformer model, $G_S$, is trained to generate a low-resolution feature grid $S$, which represents the coarse, sparse structure of the 3D object. The model $G_S$ aims to generate $S$ using an RFM objective Eq.1. In the second stage, another transformer model $G_L$, generates the detailed local latents $z_i$, conditioned on the sparse structure $p_i$, produced from the first stage. This model is also trained with an RFM objective. Conditional information, such as text embeddings, is injected into both transformers via standard cross-attention layers, while time step information is integrated using adaptive layer normalization (AdaLN). The final output is a set of structured latents, $z = (z_i, p_i)$, which combines the detailed local features with their corresponding positions in the sparse structure. This complete representation can then be rendered into a high-fidelity 3D mesh, NeRF (Mildenhall et al., 2021) or 3D-GS (Kerbl et al., 2023), via their respective decoders. Our editing framework is designed to adapt and condition this powerful two-stage generative process. For clarity, our framework diagrams primarily illustrate the first generation stage. Notably, the second stage employs the same token concatenation training strategy.

### 3.2 DATA CONSTRUCTION PIPELINE FOR 3D EDITING

A primary obstacle to advancing instruction-guided native 3D editing is the absence of a large-scale, high-quality dataset. To address this gap, we developed a systematic data construction pipeline to generate a benchmark encompassing three fundamental editing tasks: deletion, addition, and modification. As illustrated in Fig. 1, our approach combines automated, large-scale procedural generation with a rigorous manual curation process to ensure the quality of the final dataset.

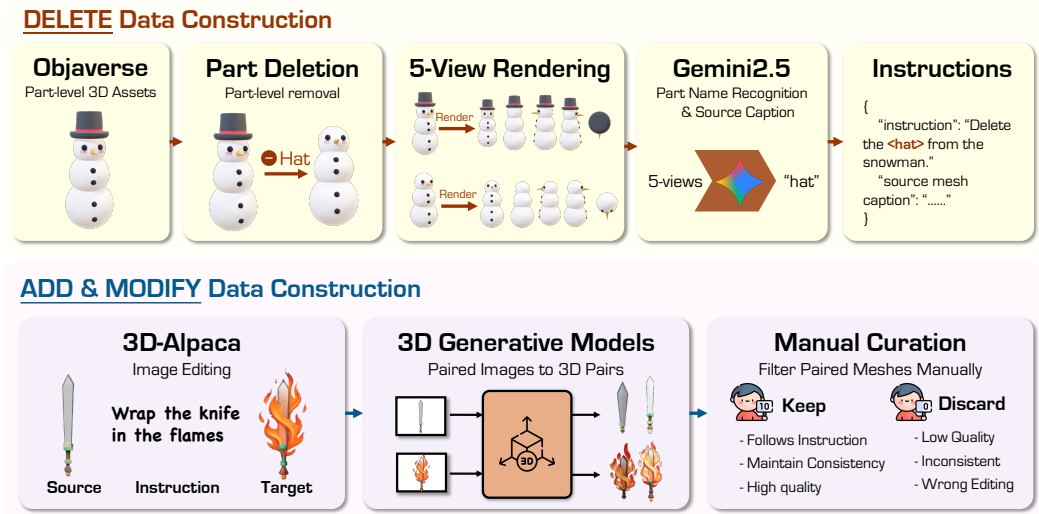

Figure 1: The overview of our data construction pipeline. For DELETE Data Construction, we programmatically remove parts from the Objaverse assets and then use Gemini 2.5 to generate instructions from multi-view renderings; it creates a source caption from views of the source object and identifies the removed part name from views of the target object. For ADD & MODIFY Data Construction, we lift 2D image pairs to 3D using a generative model, followed by a rigorous manual curation process to filter for high-quality and consistent pairs.

**DELETE Data Construction.** For the deletion task, our goal is to create source-target pairs and the corresponding instructions. We leveraged the Objaverse (Deitke et al., 2023) dataset, filtering for 3D assets with well-defined, hierarchical part structures. The generation process is as follows: Firstly, for each selected 3D asset, which serves as the source object, we programmatically identify and remove a single, distinct part to create a corresponding target object. This automated procedure enables us to efficiently generate a vast collection of (source, target) pairs for the deletion task, as illustrated in Fig. 1. To obtain a precise textual instruction for each pair, we employ a powerful Multimodal Large Language Model (MLLM), Gemini 2.5. To enable the MLLM to identify the deleted part, we first render the source object from five canonical viewpoints (front, back, left, right, and top), with the part designated for deletion highlighted in a distinct color (e.g., purple). These five views are then fed into Gemini 2.5. Through carefully designed prompts, we instruct the model to recognize and name the highlighted component (e.g., "the hat"). Additionally, we leveraged the MLLM to generate textual descriptions of the source object following the same methodology. Finally, the final instruction is formulated as a template: "delete the [part name], source 3d caption". This process yields a complete data triplet: (source object, target object, instruction), forming a multi-modal, high-quality, and high-fidelity dataset for the deletion task.

**ADD & MODIFY Data Construction.** To generate data for addition and modification tasks, we build upon existing 2D editing datasets (Ye et al., 2025). Our pipeline lifts these 2D data pairs into the 3D domain and selects high-quality, high-consistency 3D pairs through rigorous manual curation, as shown in Fig. 1. Specifically, we utilize the 3D-Alpaca dataset(Ye et al., 2025), which contains pairs of (source image, target image) and corresponding edit instructions generated by GPT-4o. We employ a state-of-the-art image-to-3D generation model, Hunyuan3D 2.1 (Hunyuan3D et al., 2025) to lift each 2D image pair into a 3D (source object, target object) pair. However, a significant challenge in the 2D-to-3D lifting process is maintaining geometric and visual consistency between the generated source and target objects. The inherent limitations of current image-to-3D models can introduce artifacts or unintended alterations. To ensure the quality of the dataset, we implement a strict manual curation protocol. Each generated triplet is evaluated against three essential criteria. Instruction Fidelity: the generated target object must accurately reflect the edit described in the instruction. Consistency of Unedited Regions: the geometry and appearance of the regions unmodified by the instruction must remain consistent between the source and target objects. Object Quality: both the source and target objects must be of high quality, free from significant artifacts,

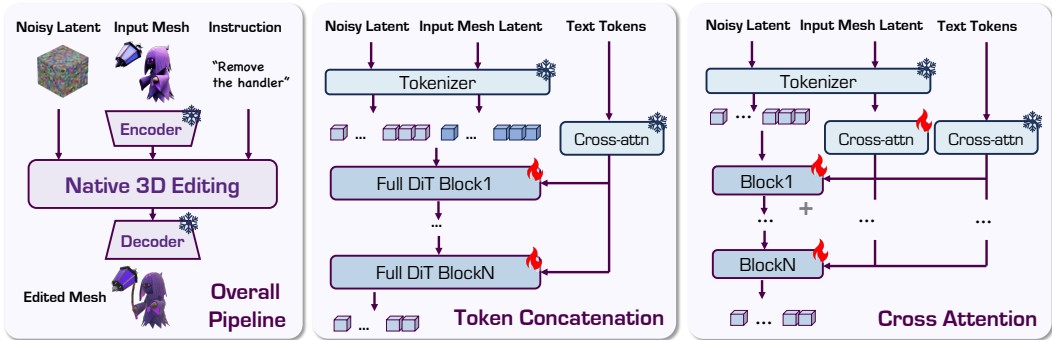

Figure 2: Overview of our proposed framework for native 3D editing. The pipeline manipulates 3D objects based on textual instructions, utilizing token concatenation as a parameter-efficient alternative to cross-attention, achieving superior editing performance without additional complexity.

broken meshes, or distorted textures. Only triplets that simultaneously satisfy all three criteria are retained. Any pair failing to meet these standards is discarded. This meticulous, human-in-the-loop filtering process is critical for constructing a reliable and high-fidelity dataset for the complex tasks of 3D object addition and modification.

### 3.3 NATIVE 3D EDITING MODEL ARCHITECTURE.

Our goal is to develop a feed-forward framework for native 3D editing that takes a source 3D object, $O_s$, and a textual instruction, $I$, as input, and synthesizes an edited target object, $O_t$, in a single, efficient pass. To achieve this, we propose harness the generative capabillity of a pre-trained text-to-3D diffusion model by imposing dual conditions $i.e.$, the source 3D object and the instruction through a meticulously designed framework. The first stage is depicted in Fig. 2. Specifically, both the source object $O_s$ and target object $O_t$ are first encoded into latent representations, $z_s$ and $z_t$, using a pre-trained 3D VAE. Concurrently, for instruction, features are extracted from the instruction $I$ using a pre-trained CLIP text encoder. These text features are then plugged into our Full-DiT model by feeding them into the cross-attention layers to guide the editing process.

**Cross-Attention strategy.** Our first attempt is to integrate the source latent and text feature conditions using a decoupled cross-attention, as illustrated in Fig. 2 (right). This strategy treats the textual guidance and the geometric reference as two sources of information, injecting them into the model through separate, parallel attention layers. Let the noisy latents of the target object at timestep $t$ be denoted by $z_t$. These latents serve as the query ($Q$) in the attention mechanism. The text instruction is encoded into an embedding $c_{text}$, and the source 3D object is represented by its token sequence $z_s$. First, the output of the cross-attention layer for the text instruction is computed as:

$$z'_t = \text{Attention}(Q, K_{text}, V_{text}) = \text{Softmax}\left(\frac{QK_{text}^\top}{\sqrt{d}}\right) V_{text}, \qquad (2)$$

where $Q = z_t W_q$, $K_{text} = c_{text} W_k$, and $V_{text} = c_{text} W_v$ are the query, key, and value matrices, respectively. Concurrently, we introduce a new, parallel cross-attention layer dedicated to the source 3D object. This layer uses the same query $Q$ but attends to the source object's tokens $z_s$:

$$z''_t = \text{Attention}(Q, K_s, V_s) = \text{Softmax}\left(\frac{QK_s^\top}{\sqrt{d}}\right) V_s, \qquad (3)$$

where $K_s = z_s W'_k$ and $V_s = z_s W'_v$. The weight matrices $W'_k$ and $W'_v$ are specific to this conditioning path. Finally, the information from both conditioning modalities is fused by simply adding their respective attention outputs. This combined representation is then integrated back into the main pathway of the transformer block:

$$z_t^{new} = z'_t + z''_t. \qquad (4)$$

This decoupled design enables the model to independently process high-level semantic guidance from the text and detailed, low-level geometric information from the source object.

Figure 3: Effectiveness of our method on deletion, addition and modification tasks. Our method facilitates precise and instruction-guided editing while maintaining the visual fidelity and structural coherence of the source object compared with other baselines.

**Token-cat strategy.** To enable a more direct and parameter-efficient integration of the source object, we propose a novel 3D token concatenation strategy. This approach, illustrated in Fig. 2 (middle), reframes the editing task as a conditional sequence-to-sequence problem, allowing the model to learn the intricate relationship between the source and target objects directly through its powerful self-attention mechanism, rather than through a separate cross-attention module.

At each training step, the source latent $z_s$ and the target noise latent $z_t$ are converted into sequences of feature vectors, $h_s$ and $h_t$ respectively. This involves patchifying the latent representations and passing them through an input projection layer, followed by the addition of positional embeddings:

$$h_s = \text{Proj}(\text{Patchify}(z_s)) + E_{pos}, \tag{5}$$

$$h_t = \text{Proj}(\text{Patchify}(z_t)) + E_{pos}, \tag{6}$$

where $z_s$ and $z_t$ are the latent representations of the source and noisy target objects. We then form a single, unified sequence, $h_{comb}$, by concatenating the source and target feature vectors along the sequence dimension:

$$h_{comb} = \text{Concat}(h_t, h_s). \tag{7}$$

This combined sequence is the primary input to our stack of Transformer blocks. Within each block, the self-attention mechanism operates on this combined sequence, allowing every token to attend to every other token. This is the crucial step where the model can directly compare and relate the features of the noisy target with the clean, stable features of the source object. The textual instruction, $c_{text}$, is still injected via a cross-attention layer within each block to guide the semantic transformation. The operation within each block can be summarized as:

$$h_{com} \leftarrow \text{FullDit}(h_{com}, c_{text}, t). \tag{8}$$

After passing through all $N$ transformer blocks, the output sequence is split to isolate the processed target structures. After passing through all $N$ transformer blocks, the output sequence is processed to produce a sparse latent representation of the target object. This concludes the first stage of our generation process, as illustrated in Fig. 2 (left).

## 4 EXPERIMENTS

In this section, we first present the experimental settings, followed by a comprehensive quantitative and qualitative comparison against state-of-the-art methodologies. We then present an ablation study evaluating the impact of various conditioning strategies and data refinement methods.

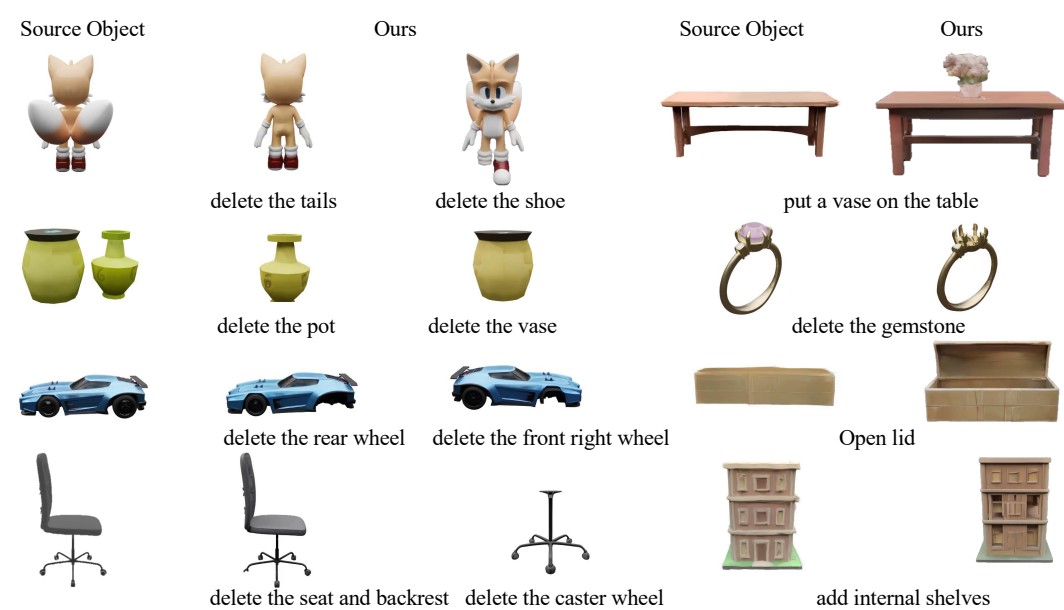

Figure 4: Experimental results on the delete, add, and modify tasks demonstrate the effectiveness of our method. The 'Source Object' and instructions show inputs, and 'Ours' displays outputs. The method excels in diverse modifications, proving its precision and versatility.

## 4.1 EXPERIMENTAL SETTINGS

**Datasets.** For training the deletion task, we select part-level 3D assets from the Objaverse dataset (Deitke et al., 2023) as source objects. By removing semantically meaningful components, we derive the corresponding target objects. We utilize Gemini 2.5 to generate accurate and descriptive text that captures the transformation between the source and the target. This process yields a curated dataset comprising 64,158 samples for the first stage and 44,890 samples for the second stage. To train the addition and modification tasks, we utilize open-source 2D editing datasets (Ye et al., 2025) that contain aligned image pairs and edit instructions, where GPT-4o generates the target images and edit instructions. Hunyuan3D 2.1 (Hunyuan3D et al., 2025) lifts these into 3D, followed by manual curation to retain only pairs with high geometric and visual fidelity. The resulting dataset contains 47,474 samples for the first stage and 47,315 for the second stage.

**Implementation Details.** Our implementation is built upon the TRELLIS architecture (Xiang et al., 2025a), serving as the backbone for both training stages. In both stages, the encoded tokens of the source object are concatenated with the input noise tensor, where the token sequence length matches the noise input's spatial dimension. To facilitate convergence, we initialize the model with pretrained weights from TRELLIS, which were originally trained on several 3D datasets comprising Objaverse (XL) (Deitke et al., 2023), ABO (Collins et al., 2022), 3DFUTURE (Fu et al., 2021), and HSSD (Khanna et al., 2024). We employ the AdamW optimizer with a fixed learning rate of 1e-4 across both training phases. In the first stage, the model is trained for 150k steps using a batch size of 12, distributed across 16 NVIDIA A800 GPUs (80GB). In the second stage, training proceeds for an additional 80k steps with a batch size of 8, across 18 A800 GPUs.

**Metrics.** Similar to previous methods (Li et al., 2025a; Barda et al., 2025), we use FID (Heusel et al., 2017) on rendered multi-view images to measure the overall visual similarity between the edited results and the original object, and use the CLIP score (Radford et al., 2021) to measure the similarity between the edited results and the edit text. FVD (Unterthiner et al., 2018) is used to evaluate the temporal continuity and consistency across multi-view images.

**Baselines.** We compare the proposed method with recent 3D editing methods, including Instant3DiT (Li et al., 2023), Tailor3D (Qi et al., 2024), TRELLIS (Xiang et al., 2025a), Hunyuan3D 2.1 (Hunyuan3D et al., 2025) and VoxHammer (Li et al., 2025a). Instant3DiT and VoxHammer

require additional 3D mask annotations, whereas Tailor3D, TRELLIS, and Hunyuan3D 2.1 depend on pre-trained 2D editing models. In contrast, our end-to-end editing framework offers a more streamlined and user-friendly alternative, eliminating the need for such auxiliary inputs.

## 4.2 Qualitative Results

To evaluate the efficacy of our method in 3D editing, we present a comparative analysis against recent state-of-the-art approaches. with a focus on delete, add, and modify operations, as visually demonstrated in Fig. 3 (a) and Fig. 3 (b) respectively. For the delete operation, we randomly select an object part for removal that was never referenced during training. For the modify operation, we apply a transformation to a randomly chosen part using an instruction that was not encountered during training, thereby testing the model's compositional generalization under unseen conditions. As shown in Fig. 3, Instant3DiT and Tailor3D suffer from visual inconsistencies and artifacts because they reconstruct 3D geometry by fusing edited multi-view 2D projections, which inherently introduces view misalignment. TRELLIS exhibits suboptimal editing performance due to the lack of feature-space alignment between the source object appearance and the target textual instruction. While Voxhammer largely maintains visual consistency with the source object, it struggles to execute precise edits according to the given instruction, a limitation inherent to its training-free strategy.

In contrast, our method excels in three aspects. First, it achieves strong instruction-following capability. Second, it preserves the original appearance of unedited regions with high visual consistency. Third, it generalizes robustly across diverse object categories and unseen instructions. These advantages arise from our 3D token concatenation strategy and large-scale training on native 3D editing data. More results are shown in Fig. 4.

Table 1: Quantitative comparison results.

| Method | FID↓ | FVD↓ | CLIP↑ |
|---|---|---|---|
| Tailor3D (Qi et al., 2024) | 296.8 | 3090.5 | 0.217 |
| Instant3DiT (Li et al., 2023) | 255.5 | 1209.8 | 0.225 |
| Voxhammer (Li et al., 2025a) | 169.6 | 594.2 | 0.230 |
| TRELLIS (Xiang et al., 2025a) | 126.2 | 365.5 | 0.238 |
| Ours | **91.9** | **286.5** | **0.249** |

## 4.3 Comparison with State-of-the-Art Methods

We compare our method against Tailor3D (Qi et al., 2024), Instant3DiT (Li et al., 2023), TREL-LIS (Xiang et al., 2025a), and VoxHammer (Li et al., 2025a) in terms of average accuracy across delete, add, and modify editing tasks. The FID Heusel et al. (2017) and FVD Unterthiner et al. (2018) measure the distributional discrepancy between rendered images and videos before and after editing, respectively. Lower values indicate better preservation of visual appearance. The CLIP Score Radford et al. (2021) quantifies the alignment between the edited content and the given textual instruction, with higher scores reflecting more accurate instruction following. As shown in Tab.1, our approach achieves the best performance on all three metrics. This demonstrates both strong instruction-following capability and high visual consistency in preserving the appearance of unedited regions. These advantages are attributed to our 3D token concatenation strategy and large-scale training on native 3D editing data, which together enable direct, text-guided manipulation of 3D representations in 3D space.

## 4.4 More Analysis and Ablation Studies

**Ablation on Conditioning Strategies.** In Section 3.3, we present two dual-conditioning strategies to activate a pre-trained text-to-3D diffusion model. The Cross-Attention strategy fuses text and source latent features via decoupled attention, while the Token-cat strategy concatenates source and target noise latents with positional encoding. To compare the efficacy of these two strategies, we conducted the ablation study using an identical backbone architecture under both conditioning schemes. The Cross-Attention strategy, which injects the source object's features via separate cross

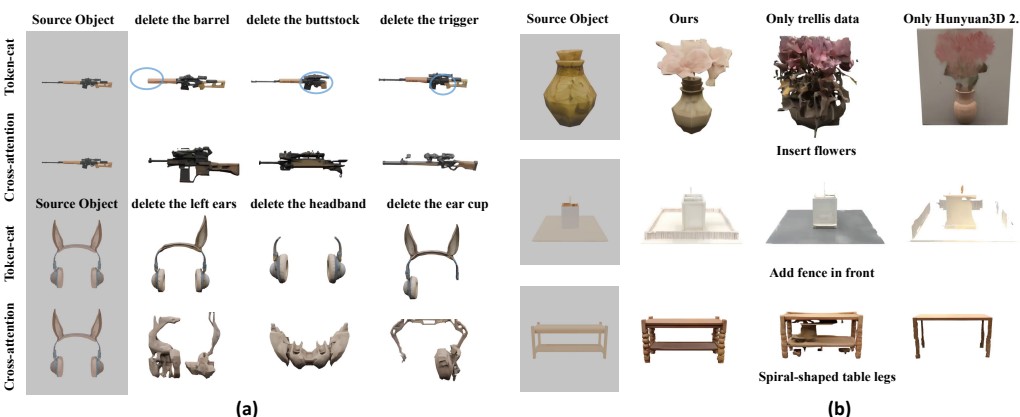

Figure 5: Ablation studies on conditioning and data refinement strategies. (a) A qualitative comparison of conditioning strategies. Our token-concatenation approach successfully performs precise edits while preserving object consistency, whereas the cross-attention method results in corrupted geometry. (b) An ablation on data sources for modification tasks. Our final model, trained on a curated dataset lifted by Hunyuan3D 2.1 ("Ours"), achieves higher fidelity than models trained on uncurated data from either TRELLIS or Hunyuan3D 2.1 alone.

attention layers, struggles to maintain consistency with the source object. As shown in the Fig. 5 (a), this approach often produces results with distorted geometry and inconsistent color, failing to execute the edit faithfully. For instance, when instructed to "delete the barrel," the model generates a completely different and corrupted rifle shape rather than performing a precise removal. In contrast, the Token-Concatenation strategy yields far superior results.

**Ablation on Data Refinement Strategies.** In Section 3.2, we introduce the data construction pipeline designed for native 3D editing. To construct a higher-quality training dataset, we conducted a comparative evaluation of different data generation tools used in our data construction pipeline. Specifically, for the addition and modification tasks, we employed both TRELLIS (Xiang et al., 2025a) and Hunyuan3D 2.1 to convert pairs of 2D images before and after editing into corresponding 3D objects. We then performed manual curation to ensure visual consistency between the original and edited versions. The filtered datasets produced by each tool were subsequently employed to train the model's addition and modification capabilities. As shown in Fig. 5 (b), models trained on the dataset synthesized by Hunyuan3D 2.1 and refined through manual curation exhibit superior performance in the 3d editing task. This improvement is attributable to the fact that Hunyuan3D 2.1 is a large-scale 3D generative model pretrained on extensive datasets, which enables it to preserve appearance consistency during 3D reconstruction better.

## 5 CONCLUSIONS

In this paper, we propose a novel native 3D editing framework that directly manipulates 3D representations in a feed-forward pass. Specifically, we create a large-scale, multi-modal dataset for instruction-guided 3D editing, covering diverse addition, deletion, and modification tasks. This dataset is meticulously curated to ensure that edited objects faithfully adhere to the instructional changes while preserving the consistency of unedited regions with the source object. Building upon this dataset, we propose a 3D token concatenation mechanism that enables parameter-efficient learning while achieving state-of-the-art performance. Comprehensive evaluations demonstrate that our approach surpasses existing multi-view editing methods, establishing new benchmarks in generation quality, 3D geometric consistency, and fidelity to user instructions.

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

# A  APPENDIX

## A.1  STATEMENT

**The Use of LLMs.**  We acknowledge the use of large language models (LLMs) for two primary purposes in this work. First, an LLM was utilized to generate the textual instructions for the editing tasks in our dataset. Second, we used an LLM to assist in refining the language of this paper, including improving grammar, phrasing, and overall clarity to ensure the content is accurate and professional.

