# OpenReview forum: "Native 3D Editing with Full Attention"
_ICLR.cc/2026/Conference — ICLR 2026 Conference Withdrawn Submission_

### Official Review · Reviewer_eqKL · 2025-10-29

**Soundness:** 1
**Presentation:** 1
**Contribution:** 1
**Rating:** 0
**Confidence:** 5

**Summary:**

This paper studies 3D scene editing. The authors first curated a new dataset of 3D editing triplets to train the model. Then, the authors repurpose the TRELLIS model with module injection and fine-tuning to build their DiT model. The proposed method outperforms the very selective baselines.

**Strengths:**

- The high-level ideas of the method are very easy to understand.
- Fig.2 is visually nice.
- A 3D editing dataset is proposed for training.

**Weaknesses:**

- **Crucial.** There is no video or multi-view presentation of the results. **My reviewing score will be 0 without these representations.**
    - Each result is only represented in a single-view image, which does not at all show clues about 3D consistency or global appearance.
    - 3D editing is a task highly dependent on qualitative results. All the quantitative metrics evaluate aspects that are only tangentially related, without considering the crucial aspects of 3D consistency and visual appearance.
    - These presentations cannot convince me at all - it is even possible that all the results are actually in 2D instead of 3D.
- **Crucial.** The baseline is very selective; all SDS-based baselines are missing. **My reviewing score will be <2 without these baselines.**
    - The only compared baselines are "Voxhammer, Instand3dit, Tailor3D, Hunyan3D-2.1, TRELLIS". All other baselines discussed in related works, including all SDS-based ones like Instruct-NeRF2NeRF, Vox-E, Tip-Editor, DreamEditor, and FocalDreamer, are not compared without a valid reason.
    - Being feed-forward does not waive the duty to compare SDS baselines, as the task is the general "3D editing", which is independent of the specific method. Similarly, a ViT-based image classification paper cannot waive the duty to compare against CNN-based classification models if those are potentially better.
- The presentation of the method is very unprofessional.
    - In Sec. 3.3, the cross-attn strategy is introduced before token-cat, but in Fig.2, token-cat is on the left, and cross-attn is on the right.
    - The formulas (2)-(8) in Sec.. 3.3 are all trivial and uninformative definitions, e.g., meaninglessly repeating the mathematical definition of attention on different groups of QKVs. Removing them will make the paper shorter without any confusion.
    - L248 "Cross-Attention strategy" is introduced as "our first attempt". This is not a suitable introduction in a formal paper, which should only appear as a "proof of work" in a course project.
    - In fact, everything about "Cross-Attention strategy", including the related ablation study, can be removed as it is not a part of the final design.
- The training dataset largely depends on an existing text-to-3D model "Hunyuan3D 2.1".
    - This makes the model somehow distilled from, and therefore bounded by, the source model Hunyuan3D 2.1, which may limit its capability.
    - Also, the generation of "a pair of 3D objects" is not mentioned using any controls to make sure the two 3D objects are similar and only differ in edited parts. Manual filtering is unable to ensure that the unedited parts are identical.
- **Crucial.** Because of the previous point, the results have many expected artifacts in "modify & add":
    - In Fig.3, "add signboard on pole" provides a completely different pole, which should be regarded as a failure.
    - In Fig.4, "put a vase on the table"'s table has a very different shape, "open lid"'s frontal textures are unexpectedly changed, and "add internal shelves" is even difficult to understand which part should be edited.
    - Even with video results, these artifacts are still not expected to be seen in the final output.
- The idea to use full attention instead of additional cross-attention for conditioning is a very common idea, instead of a novelly proposed one. Most of the modern DiT-based image and video diffusion models even compute full attention among tokens from both videos/images and texts.
- Other than TRELLIS repurposing, there are many missing necessary implementation details, including:
    - The way of "filtering for 3D assets with well-defined, hierarchical part structures".
    - The prompts to call MLLM to obtain the description of the object and the removed components.
    - The detailed criterion of the curation protocol in L210.
- There are many typos and grammatical mistakes. For example:
   - TL;DR: "instruction o **(of)** native 3d editing that directly manipulates 3D latent in 3d space without any need for multi-view editing."
   - L111. "For example, Instruct-NeRF2NeRF (Haque et al., 2023) iterative **(iteratively)** edits NeRF..."
   - Inconsistent letter cases: L248 "Cross-**A**ttention **s**trategy" v.s. L291 "Token-**c**at **s**trategy".

**Questions:**

- Could you please provide a video or multi-view (at least 4 views) results of all the edited scenes?
- Could you please compare with some state-of-the-art SDS-based methods?
- When generating triplets using Hunyuan3D 2.1, did you use any similarity control method like DiffEdit to systematically ensure that the two generated 3D scenes are identical in unedited parts?

---

### Official Review · Reviewer_EdJn · 2025-11-01

**Soundness:** 2
**Presentation:** 2
**Contribution:** 2
**Rating:** 4
**Confidence:** 4

**Summary:**

This paper proposes a novel framework for instruction-guided 3D object editing that operates natively in 3D space via a single feed-forward pass, avoiding the inconsistencies and inefficiencies of prior multi-view 2D editing pipelines. The authors create efficient data generation pipeline based on current large generative models. They explore two conditioning strategies—cross-attention and 3D token concatenation—demonstrating that the latter is more parameter-efficient and yields superior results. Some qualitative and quantitative results are provided.

**Strengths:**

- **Dataset Contribution**: The primary strength of this paper is the introduction of the large-scale dataset specifically for instruction-guided native 3D editing.
- **Extensive evaluation results**: The qualitative results shown in Figures 3 and 4 illustrate the method's superiority in maintaining consistency and following instructions.

**Weaknesses:**

- **Limited Methodological Novelty**: While the overall framework is effective, its methodological novelty is somewhat limited. The backbone architecture is an existing pre-trained model (TRELLIS). The core "novel" contribution, the 3D token concatenation strategy, is a well-established technique for conditioning in other generative domains (e.g., 2D inpainting and image editing). While the authors claim to be the first to apply this to the 3D domain, this is more of a successful adaptation than a fundamental architectural innovation.
- **Reliance on Manual Curation**: The quality of the "addition" and "modification" portions of the dataset hinges on a "rigorous manual curation process". This introduces concerns about scalability, reproducibility, and potential human biases in the filtering criteria. The paper notes that low-quality or inconsistent pairs are discarded, but it does not quantify this, making it difficult to assess the true difficulty of the 2D-to-3D lifting task or the amount of human effort required.

**Questions:**

- The ablation study in Section 4.4 and Figure 5(a) provides a clear qualitative comparison between the token concatenation and cross-attention strategies, showing the former is superior. The paper also claims token concatenation is "more parameter-efficient". Could you please quantify this claim? A small table comparing the number of additional trainable parameters required to adapt the TRELLIS backbone for the cross-attention strategy (i.e., the new attention layers ) versus the token-concatenation strategy would significantly strengthen this argument.
- Regarding the manual curation for the ADD & MODIFY dataset: What was the rejection rate? Specifically, what percentage of the initial 3D pairs generated by Hunyuan3D 2.1 were discarded for failing to meet the three criteria (instruction fidelity, consistency, and quality)? This information is crucial for understanding the dataset's construction and the current limitations of 2D-to-3D lifting models.

**Details Of Ethics Concerns:**

None.

---

### Official Review · Reviewer_ejCW · 2025-11-04

**Soundness:** 2
**Presentation:** 3
**Contribution:** 2
**Rating:** 2
**Confidence:** 4

**Summary:**

This paper proposes a native 3D editing framework that directly manipulates 3D mesh representations based on text instructions. The authors create a dataset by (1) procedurally removing parts from 3D objects for deletion tasks, and (2) lifting 2D edit pairs to 3D using image-to-3D models with manual curation for addition/modification tasks. They explore two conditioning strategies (1) cross-attention and  (2) token concatenation, finding that token concatenation performs better. The method generates edited 3D objects in 20 seconds using a two-stage transformer architecture built on TRELLIS. Experiments show improvements over baselines on FID, FVD, and CLIP metrics.

**Strengths:**

1) Large-scale dataset construction. The paper creates over 110,000 training samples covering deletion, addition, and modification tasks through a systematic pipeline.

2) Improvements on automatic metrics. Shows substantial gains over baselines (FID: 126.2 to 91.9) on standard benchmarks.

**Weaknesses:**

1) The paper claims native 3D editing avoids 2D consistency problems, but evaluates using only 2D image metrics (FID/FVD/CLIP). These metrics cannot measure 3D geometric consistency, mesh quality, or whether edits are correctly localized in 3D space.

2) No 3D geometric metrics. Missing essential measurements like Chamfer Distance on unedited regions, mesh quality checks (self-intersections, non-manifold edges), or 3D spatial accuracy of edits. Cannot verify the claimed advantage of native 3D editing.

3) No human evaluation. Automatic metrics are insufficient to assess instruction following for 3D objects. Need human judges to evaluate whether edits match instructions when viewing objects from multiple angles, whether geometry is preserved, and overall quality.

4) Incomplete ablation study. Token concatenation versus cross-attention comparison shows only 3 qualitative examples with no quantitative metrics. Cannot conclude which is actually better. Cross-attention implementation may be suboptimal.

5) Unfair baseline comparisons. Different methods use different inputs (some need masks, some use 2D editing models). Cannot isolate whether improvements come from native 3D editing, larger dataset, architecture, or other factors.


6) Missing critical details. No description of test set construction, no per-task performance breakdown (delete vs. add vs. modify), no analysis of generalization to unseen categories or instructions, no failure case discussion.


7) Minor: Limited technical novelty. Token concatenation is not new; applying it to 3D editing with a specific architecture is incremental. Main contribution is the dataset and application.

**Questions:**

1) Why no 3D geometric metrics?

2) Can you provide a quantitative comparison of token concatenation vs. cross-attention?

---

### Official Review · Reviewer_QKjY · 2025-11-11

**Soundness:** 2
**Presentation:** 3
**Contribution:** 2
**Rating:** 4
**Confidence:** 4

**Summary:**

To address the limitations that optimization-based approaches are prohibitively slow, while feedforward methods relying on multi-view 2D editing often suffer from inconsistent geometry and degraded visual quality, the authors propose a 3D editing framework that directly manipulates 3D representations in a single, efficient feed-forward pass. The work makes two main contributions:
(1) a medium-scale multi-modal dataset for instruction-guided 3D editing, and
(2) a 3D token concatenation approach as a conditional branch, whose principle closely resembles existing 2D works (e.g., CatVTON: Concatenation Is All You Need for Virtual Try-On with Diffusion Models).

**Strengths:**

1. The paper is clearly written and easy to follow.

2. The experimental comparison is thorough, benchmarking against numerous established baselines.

**Weaknesses:**

1. Limited visual quality. The method only supports very simple edits—essentially adding or replacing nearly solid-colored objects. For example, in Figure 3 (row 3), the added signboard is completely white with no texture or pattern. In contrast, HunYuan3D-2.1 produces significantly superior results. Most generated outputs also appear somewhat coarse and fall far short of the fidelity achieved by optimization-based methods (e.g., NANO3D).

2. Lack of novelty. Both the cross-attention strategy and token concatenation strategy have been extensively explored in 2D editing tasks (e.g., Animate Anyone, CatVTON). The authors merely transfer these techniques to the 3D setting without introducing meaningful adaptations or domain-specific insights for 3D representation. Consequently, the technical contribution appears incremental.

3. Missing efficiency analysis. The abstract claims that “optimization-based approaches are prohibitively slow,” yet the paper does not report inference speed or GPU memory consumption of the proposed method. Including such metrics would strengthen the evaluation.

4. The ablation study evaluates different data-generation pipelines by training models on the resulting datasets and comparing downstream performance. This indirect proxy for data quality seems unnecessary and weak. Is there no more direct way to assess the quality of the generated training data itself (e.g., via geometric consistency, visual realism, or human evaluation)?

**Questions:**

Please refer to the weakness.

---

### Note · Authors · 2025-11-14

I have read and agree with the venue's withdrawal policy on behalf of myself and my co-authors.